# Modeling Coil–Globule–Helix Transition in Polymers by Self-Interacting Random Walks

**DOI:** 10.3390/polym15183688

**Published:** 2023-09-07

**Authors:** Eddie Huang, Zhi-Jie Tan

**Affiliations:** 1Wuhan Britain-China School, No.10 Gutian Ce Rd., Qiaokou District, Wuhan 430022, China; 2School of Physics and Technology, Wuhan University, Wuhan 430074, China

**Keywords:** random walks, statistical physics, phase transition, polymer, helix

## Abstract

Random walks (RWs) have been important in statistical physics and can describe the statistical properties of various processes in physical, chemical, and biological systems. In this study, we have proposed a self-interacting random walk model in a continuous three-dimensional space, where the walker and its previous visits interact according to a realistic Lennard-Jones (*LJ*) potential $u_{LJ}\left(r\right)=\varepsilon \left[{\left(r_{0}/r\right)}^{12}-2{\left(r_{0}/r\right)}^{6}\right]$. It is revealed that the model shows a novel globule-to-helix transition in addition to the well-known coil-to-globule collapse in its trajectory when the temperature decreases. The dependence of the structural transitions on the equilibrium distance $r_{0}$ of the *LJ* potential and the temperature *T* were extensively investigated. The system showed many different structural properties, including globule–coil, helix–globule–coil, and line–coil transitions depending on the equilibrium distance $r_{0}$ when the temperature *T* increases from low to high. We also obtained a correlation form of *k*_B_*T*_c_ = *λε* for the relationship between the transition temperature *T*_c_ and the well depth ε, which is consistent with our numerical simulations. The implications of the random walk model on protein folding are also discussed. The present model provides a new way towards understanding the mechanism of helix formation in polymers like proteins.

## 1. Introduction

The random walk (RW) is a powerful model in physics and can describe the statistical properties of various processes in physical, chemical, and biological systems [1,2,3,4,5,6,7,8,9,10]. A pure random walk can be used to describe the physical Brownian motion and diffusion like the random movement of molecules/particles in liquids and gases, where the walker has no interaction with its visited sites during the random movement. Other models with interactions have also been presented to extend the categories of random walks. Self-avoiding walk (SAW) with repulsive interactions is used to describe the scale behavior of polymers in dilute solution [11], where each visited site represents a monomer of a polymer and therefore the walker cannot move to its previously visited sites. Self-attracting walk (SATW) with attracting interactions can be used to describe the structural properties of polymers in solutions [12,13]. The SATW has been shown to reveal a swelling–collapse transition at the “Θ point” *T* = Θ, a well-known phenomenon in polymer physics [14,15]. Another type of interacting walk is the active random walk model, in which the walker can change the potential of the landscape, and in turn, the changed potential will affect the behavior of the walker in the landscape afterwards [16]. In addition to the models in regular space, random walks that interact with restricted space like networks have also been extensively studied to understand the nonlinear behaviors like search and virus spreading [17]. All of these RW models have played an important role in modeling and understanding the complex phenomena in the nature [1].

Random walk models have also been an important method used for understanding the conformational changes like the coil–helix transition of polymers in a solution [14], which has been an active research area of interest for the past several decades owing to its applications related to many biological phenomena like RNA and protein folding [18,19,20,21,22,23,24,25,26]. Proteins are polymers consisting of twenty amino acid types. It has been shown that the folding reaction of the polypeptide chain undergoes a collapse transition, from a coil-like conformation to a globule-like conformation. that precedes or occurs with the final rearrangements of the protein chain to form the native structure [18]. Although tremendous progress has been made theoretically and experimentally, the mechanism of protein folding is still not fully understood [19].

As a simple model, the random walk has been a valuable tool in investigating the statistical properties of polymers in solutions [14]. However, although existing random walk models can simulate the chain collapse of coil–globule transition of polymers [12], none are able to model the coil–helix transition—an important mechanism in protein folding for helix-included proteins. Given the importance of random walks in polymer physics, a random walk model that is able to simulate the coil–helix transition will offer not only a valuable tool in understanding the mechanism of helix formation in protein folding but also a new statistical approach to the physics of polymers in general. Here, we have proposed a self-interacting random walk approach to model the conformational transition of a polymer chain. The general model, which only involves a realistic Lennard-Jones interaction between monomers, for the first time shows the coil–globule–helix transitions in random walks with the temperature decreasing. The present finding indicates that an appropriate van der Waals interaction is enough for the formation of helical structures and is expected to have a far-reaching implication in protein folding.

## 2. Materials and Method

### 2.1. Self-Interacting Walks

The self-interacting random walk (SIRW) was performed in a continuous three-dimensional space, in which the trajectory may represent a possible conformation of a polymer or polypeptide with the visited sites corresponding to the monomers or amino acids [11]. Without loss of generality, the walker starts from an initial point at $R_{0}=\left(0,0,0\right)$ which is the first visited position. At each step, the walker will randomly make a movement with a step distance of *d*_0_. Specifically, the movement of the walker is performed by randomly choosing a position among a large number (15,212 here) of uniformly distributed points on the surface of a sphere with a radius of *d*_0_ centered on the current position. The probability $P_{n\to n+1}$ of the walker moving from the *n*-th position to the (*n* + 1)-th position depends on the total potential energies of the walker at the two positions, which is expressed as [11,27],
(1)$$P_{n\to n+1}\propto e^{-\left(\frac{U_{n+1}-U_{n}}{k_{B}T}\right)}\;$$
where *k*_B_ is the Boltzmann constant and *T* is the temperature of the system. The *U_n_* is the total potential energy of the walker at the *n*-th position that depends on the previously *n* − 1 visited sites as
(2)$$U_{n}=\overset{n-1}{\underset{k=0}{\sum }}u_{k,n}\left(r\right)$$
where $u_{k,n}\left(r\right)$ is the interaction potential between the walker at the *n*-th position and a previously visited position. Here, only the van der Waals force was considered. To model a realistic interaction, we have used a Lennard-Jones (*LJ*) potential for $u_{k,n}\left(r\right)$ in the present random walk model as follows [28],
(3)$$u_{LJ}\left(r\right)=\varepsilon \left[{\left(\frac{r_{0}}{r}\right)}^{12}-2{\left(\frac{r_{0}}{r}\right)}^{6}\right]\;$$
where $r_{0}$ is the equilibrium distance at which the *LJ* interaction potential has a minimum value, ε is the depth of the potential well, and r is the distance between the *i*-th and *j*-th positions. An example of the *LJ* potential is shown in Figure 1.

Without loss of generality, the step distance *d*_0_ of each movement is set to the unit of the distance and the Boltzmann $k_{B}$ is set to 1 in the present study.

### 2.2. Numerical Simulations

According to Equations (1) and (3), our self-interacting walk model includes three parameters: the temperature *T*, the well depth of the *LJ* potential ε, and the intermolecular equilibrium distance $r_{0}$. Thus, given a set of temperature *T*, well depth ε, and intermolecular equilibrium distance $r_{0}$, the visited positions by the self-interacting walker will form a certain of structural pattern after *N* steps of movements, which would be related the folding behavior of a polymer like a protein. To investigate the structural properties of self-interacting walks, we have systematically conducted extensive numerical simulations of the self-interacting walk by changing the three parameters in the following ranges,
(4)$$\begin{array}{c} r_{0}\in \left[1.0,\;\;\;\;\;\;\;2.5\right]\\ T\in \;\left[0.001,\;100\right]\\ \varepsilon \in \;\;\left[0.5,\;\;\;\;\;\;\;\;5.0\right] \end{array}$$

We have limited the maximum number of moving steps to 50 in each simulation of our model, which is long enough to form a well-defined structure like helix in proteins. Namely, a total of 51 visited positions were considered in the system. It should be noted that each simulation run will walk an independent trajectory, though all random walk simulations are physically equivalent for the same set of parameters. As such, the image of the trajectory for any simulation run may be used as a representative for the random walks with the same set of parameters. Without loss of generality, the last run was selected for visualization in this study where the image was presented using the UCSF Chimera program [29].

### 2.3. Evaluation Metrics

Two metrics are used to measure the structural properties of self-interacting walks. One is the root-mean-square radius of gyration $R_{g}$. The radius of gyration $R_{g}$ is a commonly used parameter to characterize the overall compactness of a chain conformation and is defined as
(5)$$R_{g}^{2}=\frac{1}{N}\underset{i}{\sum }{\left(R_{i}-\bar{R}\right)}^{2}$$
where $R_{i}$ is the *i*-th visited position and $\bar{R}$ stands for the average of all visited positions.

The other metric is the helix fraction H, which is used to measure the helical property of the trajectory by the self-interacting walk and can be defined as
(6)$$H=\frac{n_{h}}{N}$$
where $n_{h}$ is the number of the visited positions forming helical structures on a trajectory of N visited positions. If all *K* continuous connections of visited positions turn in the same direction (i.e., left or right) compared to the previous connections and the sum of the *K* angles between the current and previous connections is greater than 2π, the involved *K* visited positions should form a turn of helix. Namely, the helix has *K* visited positions per helical turn and can be defined as the size of the helix. The helix fraction *H* has a value ranging from 0.0 for a random coil to 1.0 for a perfect helix. The helix size *K* could be various numbers for the helical structures in real systems like proteins. Thus, in the present study, we have chosen *K* to be 10 in order to take into account some fluctuations in our numerical simulations.

Due to the nature of random walks, the data acquired will show fluctuations in each simulation run. Therefore, we performed 1000 independent random walk simulations and calculated the statistical averages $\langle R_{g}\rangle$ and 〈H〉 for the radius of gyration and helix fraction of the system in order to remove the fluctuations.

## 3. Results and Discussion

### 3.1. Structural Properties at Low Temperature

We first investigated the dependence of the structural properties of our self-interacting walks on the intermolecular equilibrium distance $r_{0}\;$ of the *LJ* potential. To focus on the effect of the moving step distance, we have set the well depth ε to be 1.0. Then, we investigated the structural properties of our self-interacting walks with the intermolecular equilibrium distance $r_{0}$ ranging from 1.0 to 2.5 at a low temperature of *T* = 0.01.

Figure 2 shows the structures of our self-interacting walks with different intermolecular equilibrium distances. It can be seen from the figure that the self-interacting walks show interesting structural properties when the moving step distance varies, where the structures can be grouped into three broad categories: compact globule, ordered helix, and extended coil. The formation of such structures can be understood as follows. When the system has a very low temperature of *T* = 0.01, according to Equation (1), the movement will be mainly determined by the interaction energy of the walker. Therefore, when the step distance d is not large, e.g., $r_{0}=1.2$, the walker will be attracted by its previously visited positions. As such, the walker will move towards the neighboring sites of its previously visited positions and form a collapsed or compact globule structure. However, at some equilibrium distances like $r_{0}=1.5$, the walker will have the most favorable interaction energy when its visited positions satisfy a special helical geometry. As such, the self-interacting walks will lead to a helical structure at low temperatures. When the equilibrium distance is larger than the two step distances, e.g., $r_{0}=2.1$, the walker will try to move away its two previously visited positions so as to achieve the best potential energy. As such, the walker will form an extended trajectory.

### 3.2. Implications in Protein Folding

The present self-interacting walks will have a valuable implication on understanding the structures of proteins because proteins also include three basic structures of compact globule, helix, and random coil in the folded structure. In other words, the formation of the helix structure in proteins may be due to the appropriate interaction distance between the amino acids in the polypeptide. This is indeed the case in experimentally observed protein structures. The statistically average distance (i.e., equilibrium distance $r_{0}$) between two “nonbonded” C_α_ atoms is about 1.5 times the C_α_-C_α_ “bond” distance (i.e., step distance $d_{0}$) in real proteins, which is consistent with the finding of $r_{0}=1.5d_{0}$ for a helical structure in our self-interacting walks.

In addition, there are three different sizes of helical structures in real protein structures, the 3_10_ helix (*i* + 3 → *i* hydrogen bonding), the α-helix (*i* + 4 → *i* hydrogen bonding), and the π-helix (*i* + 5 → *i* hydrogen bonding), where the middle-size α-helix is the most common one [30]. Figure 3 shows the helix fraction *H* of the structures as a function of the intermolecular equilibrium distance $r_{0}$ at the temperature *T* = 0.01 and well depth *ε* = 1.0. The *y*-axis represents the helix parameter and the *x* axis represents the equilibrium distance $r_{0}$. By observation, we can see that there are three different types of structures: the globule, the helix, and the string. There are three peaks reaching 1.0, meaning there are three formations of stable helices at $r_{0}=1$.52, 1.69, and 1.82, respectively, which corresponds exactly to the number of helix types in realistic proteins. The fourth peak does not reach 1.0, so it is not a stable helix. Each type of helix has a different size, which can be expressed by the number of steps it takes per helical turn. The size of the helices from left to right are four (steps per helical turn), five (steps per helical turn), and five (steps per helical turn). The data are qualitatively consistent with the actual sizes of protein helices being the 3_10_ helix with 3.0 amino acid residues per helical turn, the α-helix with 3.6 amino acid residues per helical turn, and the π-helix with 4.1 amino acid residues per helical turn. As seen in the graph, the intervals of how long the helices are stable are also different from each other. The helix being second in size has the longest range of $r_{0}$, which is also consistent with the experimental finding that the α-helix being second in size is the most common type of protein helix.

### 3.3. Comparison with Other Models

Similar coil–helix–globule transitions have also been observed in polymer systems with an increase in self-attractive interactions [31,32] like self-attractive semiflexible ring chains [32]. Specifically, the semiflexible ring polymer chain consists of N effective monomers, where neighboring monomers are connected by the finitely extendable nonlinear elastic potential [32]. The interactions between nonbonded monomers are described by the standard Lennard-Jones potential. In addition, the stiffness of semiflexible polymers is modeled by angle-dependent bending potential between adjacent bonds. An off-lattice Monte Carlo simulation is used to investigate the conformations of the self-attractive semiflexible ring polymer [32]. Depending on the bending energy and the self-attractive interaction between monomers, the system can show a coil–helix–globule transition. It is also revealed that the transition is attributed to the competition of the configurational entropy, the bending energy, and the self-attractive interaction [32].

Although both the semiflexible polymer chains and our present self-interacting walks can exhibit a coil–helix–globule transition, our self-interacting walks do not contain a bending potential between adjacent sites compared with the semiflexible chains. It suggests that the bending potential between adjacent monomers may not be a necessary energy term for a coil–helix–globule transition in polymers. Namely, a coil–helix–globule transition can be driven by the competition of the configurational entropy and the self-attractive interaction only, as shown in our self-interacting walks. This finding will be valuable for understanding protein folding.

### 3.4. Impact of the Temperature T

#### 3.4.1. Globule–Coil Transition

We further investigated the impact of the temperature *T* on the structure properties of self-interacting walks by fixing the well depth *ε* = 1.0 and the equilibrium distance *r*_0_ = 1.0. In such cases, the self-interacting walk model does not yield a helical trajectory from low temperature of *T* = 0.001 to high temperature of *T* = 100.0, as indicated by the near-zero helix fraction of 〈H〉 in Figure 4. Nevertheless, the model shows a structure transition from the compact globule to the random coil when the temperature increases (Figure 4 and Appendix A).

Figure 5 shows the average root-mean-square radius of gyration $\langle R_{g}\rangle$ of the trajectory generated by the model as a function of temperature *T*. It can be seen that the radius of gyration $\langle R_{g}\rangle$ increases from 6.25 to 14.39, indicating a phase transition of the model from compact globule to random coil. By calculation of the average energy of the two phases, we can find that the energy difference of the two states equals ~1.0, which corresponds to the value of *k*_B_*T* when the phase transition occurs. Before the transition occurs, the interaction potential dominates the movement of the walker, attracting the walker inwards towards the previously visited positions, causing the trajectory to form a ball-like globule expanding outwards slowly. The transition occurs because the kinetic energy of the walker affected by the temperature exceeds the *LJ* interaction potential with the previous positions, causing the probability of each step to become near equal, which makes the model pure random walk. Therefore, the trajectory becomes a random coil and expands faster than a globule structure, causing the root-mean- square radius of the trajectory to be bigger (Figure 5).

#### 3.4.2. Helix–Globule–Coil Transition

We further investigated the impact of the temperature *T* on the structure properties of self-interacting walks by fixing the well depth *ε* = 1.0 and the intermolecular equilibrium distance at *r*_0_ = 1.52 because such a distance can yield a good helix at low temperature. Then, we systematically investigated the structures formed by self-interacting walks at the temperature *T* ranging from 0.001 to 100.0, where the structure properties are characterized by two parameters, average radius of gyration $R_{g}$ and helix fraction *H*.

Figure 6 shows the average helix fraction 〈H〉 of the trajectories as a function of the temperature *T* when the intermolecular equilibrium distance $r_{0}$ = 1.52 and the well depth *ε* = 1.0. It can be seen from the figure that the trajectory of the model can form a perfect helical structure with 〈H〉≈1.0 at low temperature, then turn into compact globule structure and then random coil structure both of which do not possess any helical features as the temperature increases.

Figure 7 shows the average root-mean-square radius of gyration $\langle R_{g}\rangle$ of the trajectory generated by our model as a function of temperature *T* when the equilibrium distance $r_{0}\;$= 1.52. It can be seen from the figure that the average radius of gyration $\langle R_{g}\rangle$ shows a reentrant shape with the temperature decreasing, indicating two phase transitions of the system during the process. One transition occurs around *T* = 1.25 where $\langle R_{g}\rangle$ reduces rapidly from about 16 to around 9 for the system of size *N* = 51. The $\langle R_{g}\rangle$ stays around the value for a range of temperatures until a second transition happens at temperature *T* ≈ 0.05 where the $\langle R_{g}\rangle$ jumps from around 9 up to above 20. A dramatic change of the radius of gyration $\langle R_{g}\rangle$ is an indication of a phase transition in the system. The corresponding transitions can also be confirmed by the structural change of the trajectories at different temperatures shown (Appendix A).

The temperature-dependent behavior of the system can be understood as follows. The first transition around *T* = 1.25 corresponds to the coil-globule collapse of the trajectory at the Θ point, a well-known transition in polymer physics [14] and random walk models [12]. Namely, at high temperatures, the system is dominated by the thermal fluctuation, and the trajectory has a random coil-like structure. With the temperature decreasing, the system will tend to be determined by the attractive interaction, and the trajectory will form a compact conformation. At the transition point *T* ≈ 1.25, a chain collapse occurs where the conformation changes from a coil-like phase to a globule-like phase, and the trajectory shows an intermediate conformation (Figure 7).

The second transition around *T* ≈ 0.05 is novel and has not been observed in previous random walk models. The globule–helix transition can be understood by considering a short random walk as an example. At low temperatures, one expects that the trajectory might adopt two different conformations: helix and globule, given their compact structures. For short trajectories, the number of contacts between visited positions in a helical state is expected to be comparable to that in a globule state because the structures for both states are similarly compact at low temperatures, as indicated by their comparable radii of gyration (Figure 7). Moreover, compared to the globule phase, the visited positions in the helix state are much more ordered and therefore can form more pairs of favorable interactions between each other. As a result, the ordered helix state has a more favorable potential energy than the disordered globule phase, explaining the physics of the globule-to-helix transition in our model. In other words, the helix is an optimal conformation for the trajectory of the random walk that interacts via an appropriate isotropic interaction with a minimum.

### 3.5. Diagram of Phase Transition

To further investigate the structural transition behavior of globule–helix–coil, we have systematically studied the self-interacting walks with ε ranging from 0.5 to 5.0 and *T* ranging from 0.001 to 100.0 by fixing the equilibrium distance $r_{0}\;$= 1.52. The structural properties of the trajectories easily can be tracked by their average radius of gyrations $\langle R_{g}\rangle$ and helix fraction 〈H〉. Figure 8 plots a schematic phase diagram of the system in the *T* − ε space. It can be seen from the figure that the space is divided into three regions where the upper region corresponds to the random-coil phase, the lower region corresponds to the helical phase, and a globule phase is in between.

The diagram can be understood as follows. Following the thermodynamics theory in random walks, the transition of the trajectory is determined by the balance of two competing factors: the potential energy due to attractive interactions and the thermal fluctuation. In the present model, the potential energy due to the attractive interaction can be characterized by the potential well depth *ε*, and that due to the thermal fluctuation can be quantified in terms of *k*_B_*T*. Therefore, the transition temperatures *T*_c_ as a function of *ε* may be assumed to have a form as
(7)$$k_{B}T_{c}\approx \lambda \varepsilon \;\;$$
where *λ* is an empirical parameter depending only on the conformational change between two phases.

For the coil-to-globule collapse that corresponds to the first transition here, the system changes from a random coil state where the walker can move freely with an effective interaction potential of zero to a compact state where the effective interaction is expected to be on the order of the potential well depth ε. Considering that the walker can interact with more than one previous visit during the random walk, the parameter *λ* is expected to be >1.0. As shown in Figure 8, Equation (7) is consistent with the simulation results when *λ* = 1.25. For the globule-to-helix transition, the thermal fluctuation effect will approach zero at low temperatures. However, the potential energy for the walker is more favorable in an ordered helix state than in a disordered globule state, as discussed above. Since the globule–helix transition is expected to be the same from the conformational change point of view for different potential well depths, *λ* is expected to have the same value for different *ε*. As shown in Figure 8, this is indeed the case and when *λ* = 0.05, Equation (7) is in excellent consistency with the transition temperatures obtained by our simulations.

### 3.6. Structure Transitions at Other Equilibrium Distances

Furthermore, we have also investigated the structures of the trajectories at other equilibrium distances of the *LJ* potentials, including $r_{0}\;$= 1.69, 1.82, and 2.25. Similar to the case for $r_{0}\;$= 1.52 (Appendix A), the models also exhibit a helix–globule–coil transition for the cases of $r_{0}\;$= 1.69 (Appendix A) and $r_{0}\;$= 1.82 (Appendix A). However, for the case of $r_{0}\;$= 2.25 (Appendix A), the model only exhibits an extended-string–random-coil transition, as expected.

## 4. Conclusions

To conclude, we have presented a novel self-interacting random walk model, in which the interaction between the walker and its previous visits is described by a realistic van der Waals interaction using a Lennard-Jones potential. Among existing random walks, our model shows a new globule–helix transition in addition to the well-known coil-globule collapse when the temperature decreases. The dependence of the transitions on the temperature *T* and the well depth of the potential *ε* were investigated, and the relationship between the transition temperature *T*_c_ and the well depth of the potential *ε* was derived. The present model might provide a method towards understanding the physics and mechanism of protein folding because the thermodynamic behavior of the system was found to be consistent with the folding transition in the hydrophobic collapse model of protein folding.

## Figures and Tables

**Figure 1 polymers-15-03688-f001:**
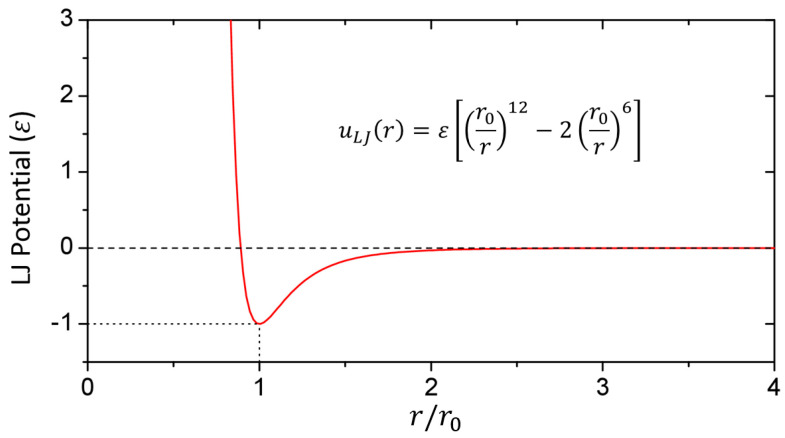
A graph of the Lennard Jones potential function, where the function has a minimum potential of −ε at the intermolecular equilibrium distance $r_{0}$.

**Figure 2 polymers-15-03688-f002:**
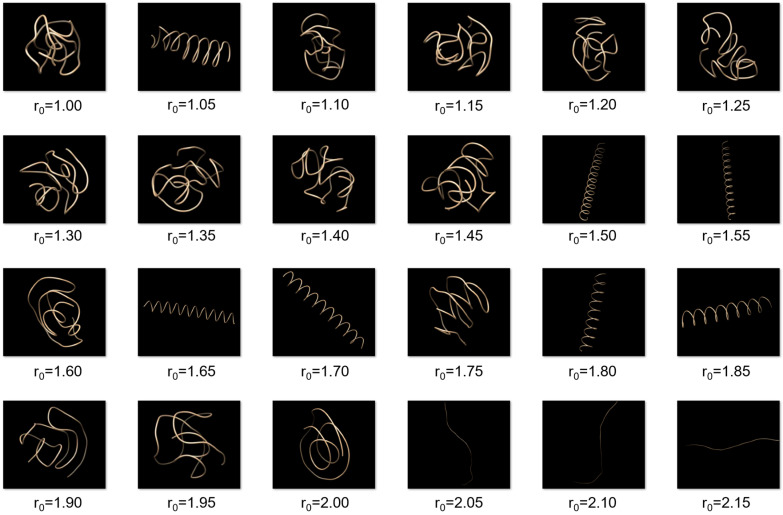
Structures of the typical trajectories by self-interacting walks at different intermolecular equilibrium distance $r_{0}$ of the *LJ* potential when the temperature *T* = 0.01 and well depth *ε* = 1.0.

**Figure 3 polymers-15-03688-f003:**
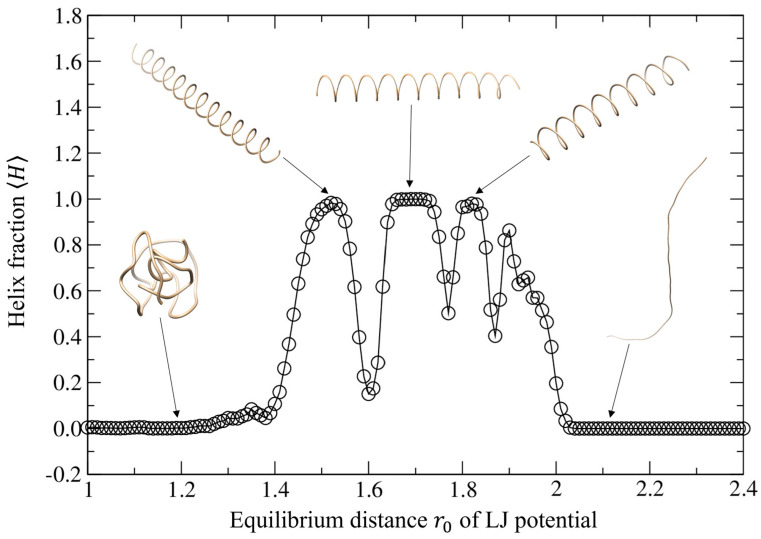
Average helix fraction 〈H〉 of the trajectories for different intermolecular equilibrium distances $r_{0}$ when the temperature *T* = 0.01 and well depth *ε* = 1.0, where several typical trajectories are shown for the corresponding equilibrium distances.

**Figure 4 polymers-15-03688-f004:**
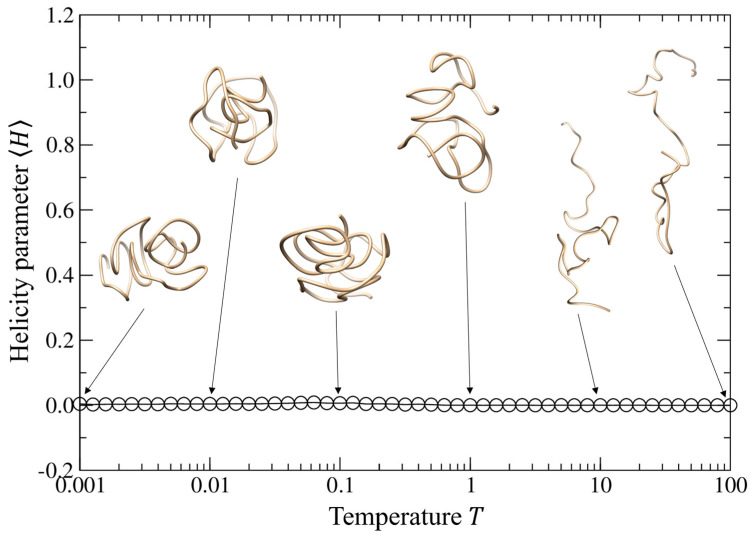
Average helix fraction 〈H〉 of the trajectories as a function of the temperature when the intermolecular equilibrium distance $r_{0}\;$= 1.0 when and the well depth *ε* = 1.0, where several typical trajectories are shown for the corresponding equilibrium distances.

**Figure 5 polymers-15-03688-f005:**
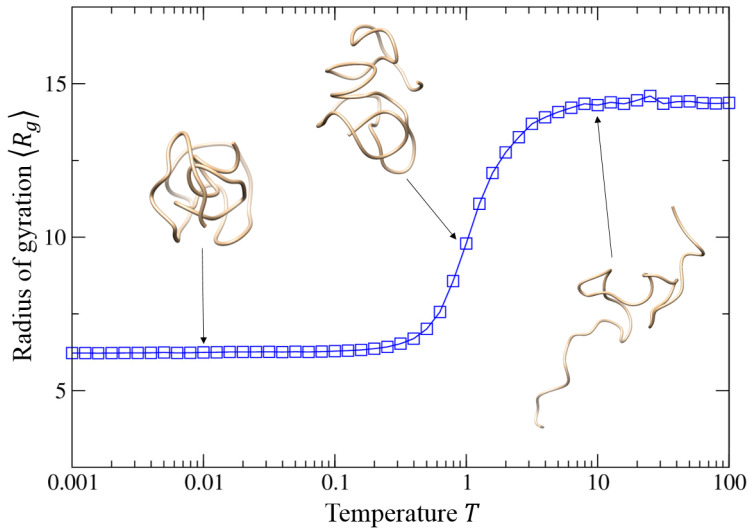
Average radius of gyration $\langle R_{g}\rangle$ of the trajectories as a function of the temperature when the intermolecular equilibrium distance $r_{0}\;$= 1.0 and the well depth *ε* = 1.0, where three typical trajectories are shown for the corresponding equilibrium distances.

**Figure 6 polymers-15-03688-f006:**
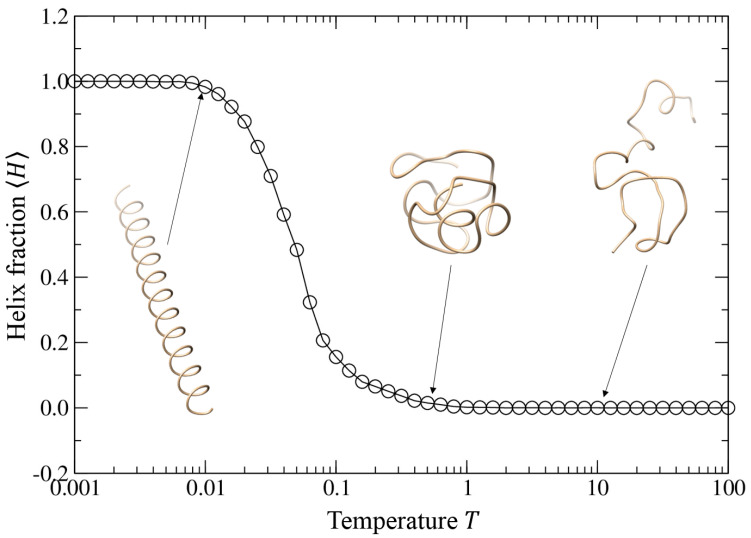
Average helix fraction 〈H〉 of the trajectories as a function of the temperature when the intermolecular equilibrium distance $r_{0}\;$= 1.52 when and the well depth *ε* = 1.0, where several typical trajectories are shown for the corresponding equilibrium distances.

**Figure 7 polymers-15-03688-f007:**
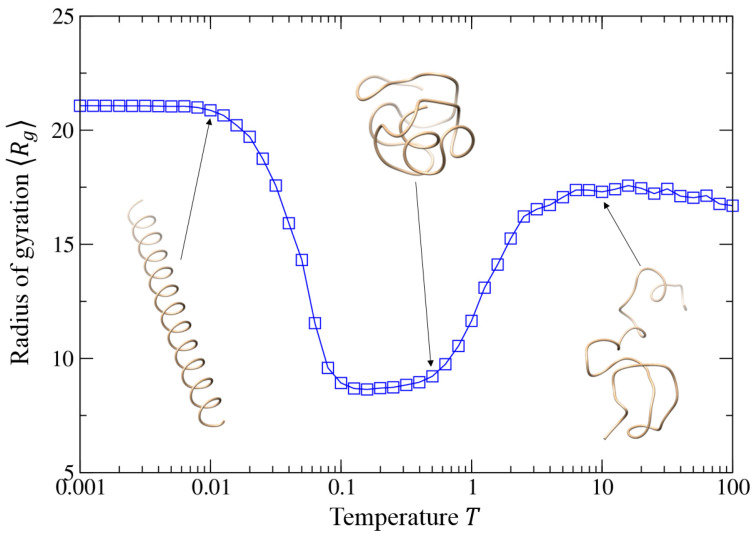
Average radius of gyration $\langle R_{g}\rangle$ of the trajectories as a function of the temperature when the intermolecular equilibrium distance $r_{0}\;$= 1.52 and the well depth *ε* = 1.0, where three typical trajectories are shown for the corresponding equilibrium distances.

**Figure 8 polymers-15-03688-f008:**
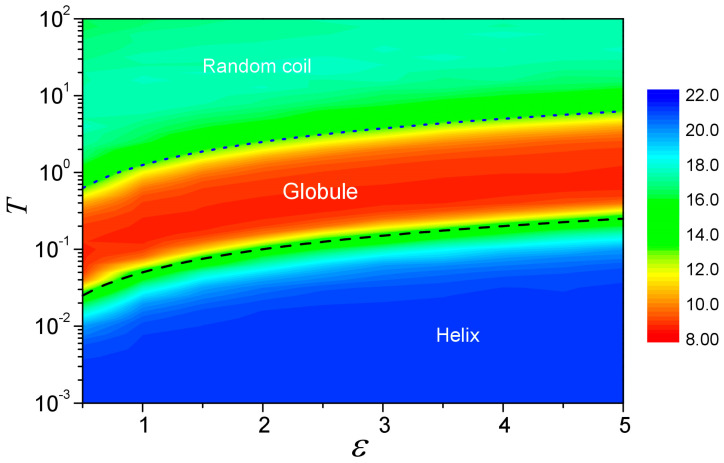
A schematic phase diagram of the system in the *T* − ε space when for a fixed *r*_0_ = 1.52. The system size is *N* = 51. The diagram is colored from red to blue based on the radius of gyration $\langle R_{g}\rangle$. The dotted and dashed lines stand for the plots of Equation (7) with *λ* = 1.25 and 0.05, respectively.

## Data Availability

The data presented in this study are available upon request from the corresponding authors.

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
