# Peer review of "Modeling Coil–Globule–Helix Transition in Polymers by Self-Interacting Random Walks"

_polymers, 2023, doi:10.3390/polym15183688_

Round 1

Reviewer 1 Report

Please read the attached file for details.

Please read the attached file for details.

Reviewer 2 Report

In this manuscript, by Huang and Tan, the authors present a self-interacting random walk (SITW) model to study the structural properties of polymers. The interaction between the steps in this random walk are described by the Lennard-Jones potential. Structures resulting from the SITW model are studied covering a wide range of three parameters that characterize the model - step length (via equilibrium distance), strength of the interaction, and temperature. Structures observed for different step lengths are consistent with the protein structures found in nature. The proposed model shows a new globule-helix transition in addition to the coil-globule structural transition on lowering the temperature. A relation between the transition temperature and the strength of the interaction is obtained. The work is original and of interest. The manuscript has been written quite well with a clear logical flow. As such I do not have any hesitation in recommending this manuscript for publication in 'polymers'. I just have some comments listed below that the authors may address before publication. 

1. The authors state that they have simulated 1000 random walks for each setting of the parameters, but the image of the trajectory for the last run is plotted for visualization. How do the preceeding 999 simulations inform the last simulation? Do they influence the outcome of this simulation in any way? If they do not, are statistical averages calculated for some properties over these 1000 simulations? What properties are these? If the last simulation is independent of any previous simulation and no statistical averages are calculated, what is the use of carrying out so many simulations instead of just one?

2. A very minor proof level comment is that the authors quote a slightly different value of r0 at lines 229 (1.5) and 234 (1.52). Which of the two is correct?

Round 2

Reviewer 1 Report

The revised manuscript is good for publication.